# Analysis of fast calcium dynamics of honey bee olfactory coding

Marco Paoli[1,2]*, Antoine Wystrach[2], Brice Ronsin[3], Martin Giurfa[1,2,4]*

[1]Neuroscience Paris-Seine – Institut de biologie Paris-Seine, Sorbonne Université, INSERM, CNRS, Paris, France; [2]Centre de Recherches sur la Cognition Animale, Centre de Biologie Intégrative, Université Paul Sabatier, CNRS, Toulouse, France; [3]Centre de Biologie Intégrative, Université Paul Sabatier, CNRS, Toulouse, France; [4]Institut Universitaire de France (IUF), Paris, France

**Abstract** Odour processing exhibits multiple parallels between vertebrate and invertebrate olfactory systems. Insects, in particular, have emerged as relevant models for olfactory studies because of the tractability of their olfactory circuits. Here, we used fast calcium imaging to track the activity of projection neurons in the honey bee antennal lobe (AL) during olfactory stimulation at high temporal resolution. We observed a heterogeneity of response profiles and an abundance of inhibitory activities, resulting in various response latencies and stimulus-specific post-odour neural signatures. Recorded calcium signals were fed to a mushroom body (MB) model constructed implementing the fundamental features of connectivity between olfactory projection neurons, Kenyon cells (KC), and MB output neurons (MBON). The model accounts for the increase of odorant discrimination in the MB compared to the AL and reveals the recruitment of two distinct KC populations that represent odorants and their aftersmell as two separate but temporally coherent neural objects. Finally, we showed that the learning-induced modulation of KC-to-MBON synapses can explain both the variations in associative learning scores across different conditioning protocols used in bees and the bees' response latency. Thus, it provides a simple explanation of how the time contingency between the stimulus and the reward can be encoded without the need for time tracking. This study broadens our understanding of olfactory coding and learning in honey bees. It demonstrates that a model based on simple MB connectivity rules and fed with real physiological data can explain fundamental aspects of odour processing and associative learning.

*For correspondence:
marco.paoli@sorbonne-universite.fr (MP);
martin.giurfa@sorbonne-universite.fr (MG)

**Competing interest:** The authors declare that no competing interests exist.

## eLife assessment

How neural circuits represent sensory signals during and after stimulus presentation is a central question in neuroscience. Here, a model of the insect mushroom body, constructed from simple, known synaptic connectivity rules, is shown to **convincingly** explain stimulus discrimination and associative memory, even in the presence of variability in the input signals as experimentally measured from the antennal lobe of the honeybee. This **important** study makes testable predictions for the role of specific neurons in a neural circuit for associative memory, of relevance to any study of neural network design and operation.

## Introduction

The logic of olfactory coding and learning has been extensively studied at the behavioural and neural levels in various insect models (*Adam et al., 2022*; *Galizia, 2014*; *Giurfa, 2015*; *Jefferis et al., 2007*). Among them, the honey bee *Apis mellifera* has played a pivotal role in our understanding of these processes due to its behavioural accessibility and tractability of its nervous system. In bees, olfactory

perception and learning have been typically investigated using the proboscis extension reflex (PER; *Giurfa, 2007*; *Menzel, 1999*), a protocol that relies on pairing a neutral olfactory stimulus (the conditioned stimulus or CS) with a positive reinforcement of sugar solution (the unconditioned stimulus or US) (*Bitterman et al., 1983*; *Giurfa and Sandoz, 2012*; *Takeda, 1961*). In the classical version of PER conditioning, the CS precedes and partially overlaps US presentation. This results in high levels of specific memory for the conditioned odorant. Conversely, when the reward precedes the stimulus (backward conditioning), no positive association can be established (*Felsenberg et al., 2014*; *Hellstern et al., 1998*). It also enables the study of more sophisticated cognitive processes such as trace learning (*Paoli et al., 2023a*; *Szyszka et al., 2011*) and patterning discrimination (*Deisig et al., 2001*; *Devaud et al., 2015*). In the former, CS and US are not overlapping but separated by a stimulus-free temporal gap, whereas in the latter, bees are trained to respond in opposite ways to a two-odorant mixture compared to its individual components (*Deisig et al., 2001*; *Devaud et al., 2015*). Overall, the olfactory conditioning of PER provides a robust read-out for investigating the dynamics of olfactory memory formation (*Giurfa and Sandoz, 2012*; *Villar et al., 2020*) and olfactory perception (*Guerrieri et al., 2005*).

Our understanding of the neural processing subtending olfactory coding and learning relies on decades of neuroanatomical and neurophysiological studies performed in honey bees (*Paoli and Galizia, 2021*). Olfactory processing starts at the peripheral level, when volatile chemicals interact with the olfactory receptors expressed on the dendritic membrane of the olfactory sensory neurons (OSNs). The biochemical nature of odorant-receptor interactions allows for a certain molecule to bind to multiple receptors with different affinities, resulting in the activation of an OSNs sub-population with odorant-specific response intensities and latencies (*Münch and Galizia, 2016*). Olfactory sensory neurons innervate the first olfactory processing centre, the antennal lobe (AL), where the neurons expressing the same olfactory receptor converge onto one of ~160 glomeruli, the anatomical and functional units of the AL (*Flanagan and Mercer, 1989*). Thus, odorant detection results in the stimulus-specific activation of a subset of glomeruli, creating a stereotypical map of odour-induced glomerular responses (*Galizia et al., 1999*; *Sachse et al., 1999*). The signal processed in the AL is forwarded by ~800 output neurons – the projection neurons (PNs) – to higher order brain centres: the mushroom bodies (MBs), central, paired structures dedicated to multisensory integration, memory storage and retrieval (*Heisenberg, 2003*; *Stopfer, 2014*), and the lateral horn of the protocerebrum, a diffuse bilateral structure involved in valence coding of odorants (*Jeanne et al., 2018*; *Roussel et al., 2014*; *Strutz et al., 2014*). The MB architecture is defined by the layout of its ~185,000 intrinsic neurons, the Kenyon cells (KCs). Each KC extends its dendritic arborisation within the MB input regions, termed the calyces, where it receives input from multiple PNs, and projects its axon to the MB output region, termed the pedunculus, where it bifurcates into the vertical (α and γ) and the medial (β) lobes (*Mobbs, 1982*; *Strausfeld, 2002*). Kenyon cells integrate, among others, the excitatory input of olfactory PNs and the inhibitory input of recurrent GABAergic feedback neurons (*Ganeshina and Menzel, 2001*; *Grünewald, 1999*; *Rybak and Menzel, 1993*; *Zwaka et al., 2018*). The latter plays a critical role in shaping KCs olfactory responsiveness and maintaining a sparse output over a wide range of odorants and concentrations (*Papadopoulou et al., 2011*; *Stopfer, 2014*). Antennal lobe, mushroom body and lateral horn are innervated by the VUMmx1 neuron (*Hammer, 1993*), an unpaired octopaminergic neuron conveying appetitive reward-related information to the olfactory circuit and mediating reward-based olfactory memory formation. The activity in the Kenyon cells is integrated by MB output neurons (MBONs). These neurons exhibit which show learning-related plasticity and provide valence-loaded information to pre-motor areas (*Aso et al., 2014b*; *Okada et al., 2007*; *Schmalz et al., 2022*; *Strube-Bloss and Rössler, 2018*).

The neural representation of odorants in the AL has been extensively described by means of functional calcium imaging (*Paoli and Haase, 2018b*). In vivo imaging allowed observing that each stimulus is represented with a specific pattern of excitatory and inhibitory responses across the glomeruli of the AL (*Galizia et al., 1999*; *Sachse and Galizia, 2002*; *Sachse et al., 1999*), and that perceptually similar odorants elicit similar glomerular response patterns (*Guerrieri et al., 2005*). Calcium imaging of AL activity has been typically conducted at low temporal resolutions (~100–200ms; *Locatelli et al., 2016*; *Mertes et al., 2021*; *Nouvian et al., 2018*; *Sachse and Galizia, 2003*). This is partially justified by the slow dynamics of fluorescent calcium sensors, which resulted – in general – in the compression of olfactory representation into a spatial vector of glomerular response intensity, with a concomitant loss

of information on neural response dynamics. While calcium signal decay is relatively slow (>100ms), its onset is fast (<10ms) (*Helassa et al., 2015*; *Moreaux and Laurent, 2007*), and provides the possibility for investigating parameters such as glomerular response latency (*Junek et al., 2010*; *Paoli et al., 2018a*), signal frequency components (*Paoli et al., 2016*) and interglomerular information transfer (*Chen et al., 2023*; *Paoli et al., 2023b*). In this study, we significantly improved the temporal resolution of calcium imaging recordings of olfactory neurons in the bee brain by means of a resonant scanning multiphoton microscope. This allowed us to record calcium activity at a<10 ms resolution, preserving temporal information of the olfactory code and yielding a more realistic representation of odour trajectories in the AL. First, we observed that odour representation changes dynamically during and after an olfactory stimulation, resulting in specific odour and after-odour images. Then, we showed that most glomerular response profiles presented an inhibitory component, suggesting that glomerular activity is strongly shaped by local inhibition, with only a minor influence – if any – of local excitatory interneurons.

Calcium imaging analysis was combined with a modelling approach to investigate how odorant representation evolves from the AL to the MB. Electrophysiology experiments have provided insight on the phasic and sparse activity of Kenyon cells as well as into its oscillatory nature (*Laurent and Davidowitz, 1994*; *Perez-Orive et al., 2002*; *Stopfer, 2014*). However, the unavailability of an imaging method allowing the visualisation of olfactory coding in a KC ensemble prevented us from further understanding how odorants are represented within the MB, for example how the neural representation of an odorant is transformed from the AL to the MB or to what extent the similarity among odorant representation is maintained in the KC space. One way to address these questions is by using neural network models constructed *via* the abstraction of common features of MBs across insect species (e.g. fruit fly, locust, honey bee) to reproduce cognitive tasks such as stimulus discrimination or learning. Models can be built by simulating multiple neurons interacting with each other according to physiological rules such as the dynamics of action potentials or Hebbian synaptic plasticity (*Eschbach et al., 2020*; *Finelli et al., 2008*; *Gkanias et al., 2022*; *Huerta et al., 2004*; *Smith et al., 2008*). Other models simulate the MB neural network by considering the statistics of connectivity within the neuropil, for example the ratio of PNs to KCs, the average number of synaptic connections, the neuronal firing rate (*Ardin et al., 2016*; *Buehlmann et al., 2020*; *Le Moël et al., 2019*; *Peng and Chittka, 2017*; *Springer and Nawrot, 2021*; *Wystrach, 2023*). Here, we followed the second approach and built a simplified but realistic neural network model of the MB based on the neuroanatomical and functional properties of the insect's olfactory circuit. The model comprises three layers of neurons: (1) a MB input layer to provide the model experimentally acquired time-series of PN activity; (2) a MB intrinsic layer, where the input signal is distributed to a population of modelled KCs based on neuroanatomical and physiological data; (3) a MB output layer, where one appetitive MB output neuron (MBON) receives input from the KC layer and can be subject to learning-induced plasticity. Generally, the input for this type of model is simulated based on the spatial and temporal statistics of the odour-induced glomerular activity (*Eschbach et al., 2020*; *Finelli et al., 2008*; *Gkanias et al., 2022*; *Huerta et al., 2004*; *Le Moël and Wystrach, 2020*; *Peng and Chittka, 2017*; *Smith et al., 2008*; *Springer and Nawrot, 2021*). In this case, we fed the model with the time series of the PN responses recorded via in vivo calcium imaging analysis from multiple individuals exposed to three different odorants at the MB working frequency of 20 Hz (*Cassenaer and Laurent, 2007*; *Laurent and Naraghi, 1994*).

Here, we show that an MB neural network model based solely on three simple connectivity rules abstracted from insect studies – sparse connectivity, feedback inhibition, and learning-induced synaptic modulation – is sufficient to explain fundamental MB processing features such as improved olfactory discrimination and associative learning. The model also predicts that the presence of post-stimulus inhibitory and excitatory activity leads to the recruitment of a second pool of KC, resulting in a distinct after-odour representation in the MB. Additionally, the model's performance in appetitive learning and across-stimuli generalisation was coherent with empirical measurements obtained from behavioural protocols. Finally, using real physiological datasets as input, we showed that such a model is robust to noise and biological variability across multiple stimulus repetitions.

## Results
### Response dynamics of projection neurons in high-temporal resolution

Antennal lobe PNs of eight bees were back-filled with the calcium sensor Fura-2 (*Paoli and Haase, 2018b*; *Sachse and Galizia, 2002*) to measure odour-induced glomerular activity of (~25 glomeruli/

AL) (*Figure 1A*). Calcium signal was recorded upon stimulation with two monomolecular odorants (1-hexanol, 1-heptanol) and a complex fragrance (peppermint oil) on a 5 s/25 s ON/OFF conformation for 20 trials (*Figure 1B–D*). Fast calcium imaging revealed the high variability of odorant-elicited glomerular response profiles.

First, we assessed the stability of odorant representations across time and trials given the duration of each imaging session (30 min) and the multiple stimulus repetitions. For each individual, we calculated the Pearson's correlation coefficients across pairs of trials of the glomerular response vectors before, during and after olfactory stimulation. *Figure 1E, E'* show that across-trial correlation is elevated during odour arrival and remains around 0.5seconds after stimulus termination. Moreover, a visual inspection of glomerular response profile dynamics confirmed that, with our stimulation protocol, the temporal activation and inactivation of each glomerulus appear to be conserved across stimulus repetitions (*Figure 1—figure supplement 1*). This shows that, in our experimental conditions, odour coding was stable throughout the entire imaging period.

All odorant response traces were pooled together (*Figure 1F*) and clustered according to their direction with respect to the pre-stimulus baseline (i.e. excitatory or inhibitory) and duration (*Figure 2*, see Methods), allowing for a statistical description of glomerular responses across individuals and odorants, and for the identification of classes of recurrent profiles (*Figure 2A and B*). The most common profiles comprised inhibitory (group 1) and excitatory (group 4) responses lasting the entire olfactory stimulations and terminating after odour offset. In some cases, they could result in prolonged inhibition (group 2) or excitation (group 6) lasting after stimulus offset or could be followed by a post-stimulus excitatory (group 3) or inhibitory activity (group 5). Approximately 15% of responses consisted of short phasic excitation followed by prompt signal termination (group 7) or even by an inhibitory response (group 8). About 15% of all recorded glomerular traces showed no detectable odorant-induced activity (group 9). Of all traces, 48% were excitatory, 37% inhibitory, and 15% unresponsive (*Figure 2C and D*). Moreover, 251 glomerular responses out of 546 (46%) displayed an inhibitory component either during or after olfactory stimulation. Notably, the varied response types were evenly distributed across the different combinations of bees and odorants, indicating an absence of individual bias (*Figure 2—figure supplement 1*).

The heterogeneity of PNs response profiles could be due to the diversified temporal pattern of the ONSs input (*Kim et al., 2023*) and to second-order processing mediated by lateral connections within the AL (*Girardin et al., 2013*; *Krofczik et al., 2008*). We assessed the contribution of these two components by measuring the latency of excitatory and inhibitory glomerular responses (i.e., the timepoint $t$ at which a response profile exceeds the threshold of one standard deviation from the mean pre-stimulus activity). *Figure 2E* shows that excitatory responses (groups 4–8) all share a similar onset (313±22ms, n=263), which is shorter than that of inhibitory profiles (groups 1–3; 351±47ms, *n*=201) (Kruskal-Wallis test, p<0.05, Tukey-Kramer multiple comparison correction). Moreover, the latency of short excitatory responses' termination (groups 7 and 8; 346±47ms, *n*=60) is coherent with the onset of inhibitory profiles. These findings indicate that odour representation in the AL is shaped initially by the excitatory input delivered by the OSNs and reshaped – approximately 40ms later – by local inhibition. Such an olfactory tuning is different from what is proposed in *Drosophila*, where both excitatory and inhibitory local neurons contribute to moulding the neural correlate of an olfactory input (*Chou et al., 2010*; *Olsen et al., 2007*).

## A mushroom body neural network model: Key principles

We constructed a simple but realistic MB neural network model based on the known connectivity of this structure in the insect brain. The model architecture relies on three main principles (*Figure 3*): First, in each brain hemisphere, ~800 AL projection neurons (PNs) diverge onto ~185,000 KCs (*Strausfeld, 2002*). Neuroanatomical studies in bees (*Szyszka et al., 2005*) and flies (*Litwin-Kumar et al., 2017*; *Caron et al., 2013*) suggest that each KC is randomly innervated by approximately 7–10 PNs. Thus, we generated a MB network where ~ 25 PNs (i.e., the average number of glomeruli imaged during a calcium imaging experiment) diverge onto 1000 KCs, with each KC being innervated by ~8 PNs. Second, recurrent inhibitory neurons such as the A3 feedback neurons in the bee (*Mobbs, 1982*; *Rybak and Menzel, 1993*; *Szyszka et al., 2005*; *Zwaka et al., 2018*), the APL neuron in fly (*Lin et al., 2014*), and the giant GABAergic neuron in the locust (*Papadopoulou et al., 2011*), modulate MB KCs firing rate so that less than 20% of KCs are active upon olfactory stimulation, and only ~5% are

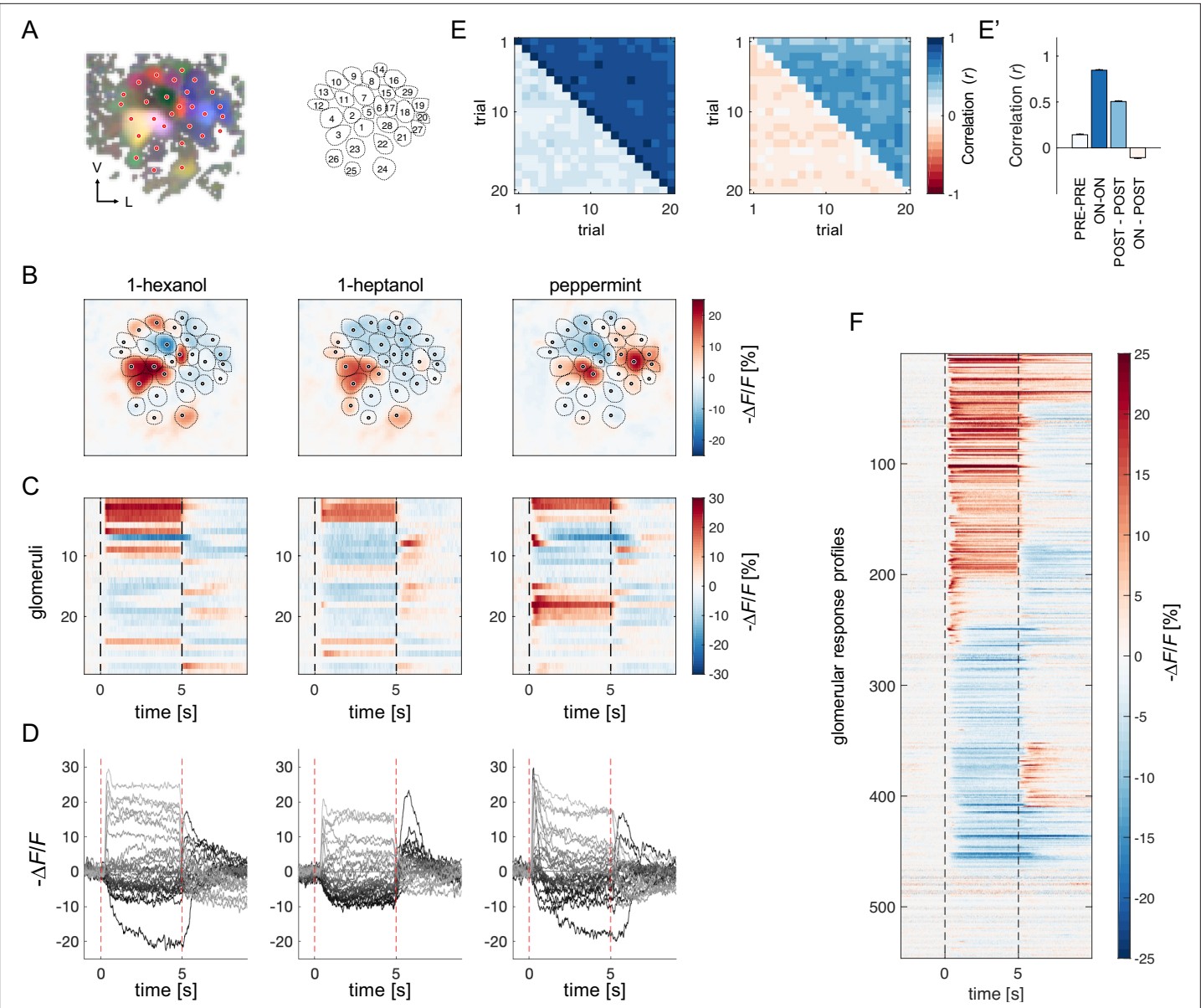

**Figure 1.** Projection neurons calcium imaging analysis. (**A**) For each AL, the odour response maps for the three stimuli were merged into an RGB image to highlight glomerular structures: 1-hexanol was set as the red channel, 1-heptanol as the green, and peppermint as the blue. Glomeruli were hand-labelled for time-series extraction. V, ventral; L, lateral. (**B**) Exemplary AL odorant responses during olfactory stimulation with three odorants. Colour bar indicates the relative change of activity during olfactory stimulation with respect to the pre-stimulus baseline. Circular areas indicate identified glomeruli according to (**A**). (**C, D**) Temporal profiles of the glomerular regions identified in (**A**) for the three odorants. Each line in (**C**), labelled from 1 to 28, refers to the glomerular ID in (**A**). Temporal profiles represent the average activity of 20 stimulations. Dashed lines in (**C, D**) limit the begin and the end of the olfactory stimulation. (**E, E'**) Pearson's correlations between pairs of glomerular response vectors across repetitions. (left) The upper right part of the matrix shows correlation scores among glomerular activity during olfactory stimulation ($t$=1–5 s after odorant onset, ON $vs$ ON) across trials; the bottom left part shows correlation scores among glomerular activity before stimulation ($t = $ –1–0 s, PRE $vs$ PRE) across trials. (right) The upper right part of the matrix shows correlation scores among glomerular activity after olfactory stimulation ($t$=1–4 s after odorant offset, POST $vs$ POST) across trials; the bottom left part shows correlation scores among glomerular activity during and after stimulation (ON $vs$ POST) across trials. (**E'**) Mean (± s.e.m.) across-trial correlation of the four combinations presented in the matrices in (**E**). (**F**) All glomerular responses from eight ALs to three odorants were pooled together to provide an overview of the complexity of response profiles ($n$=546).

The online version of this article includes the following figure supplement(s) for figure 1:

**Figure supplement 1.** Sample of glomerular response profile dynamics across trials.

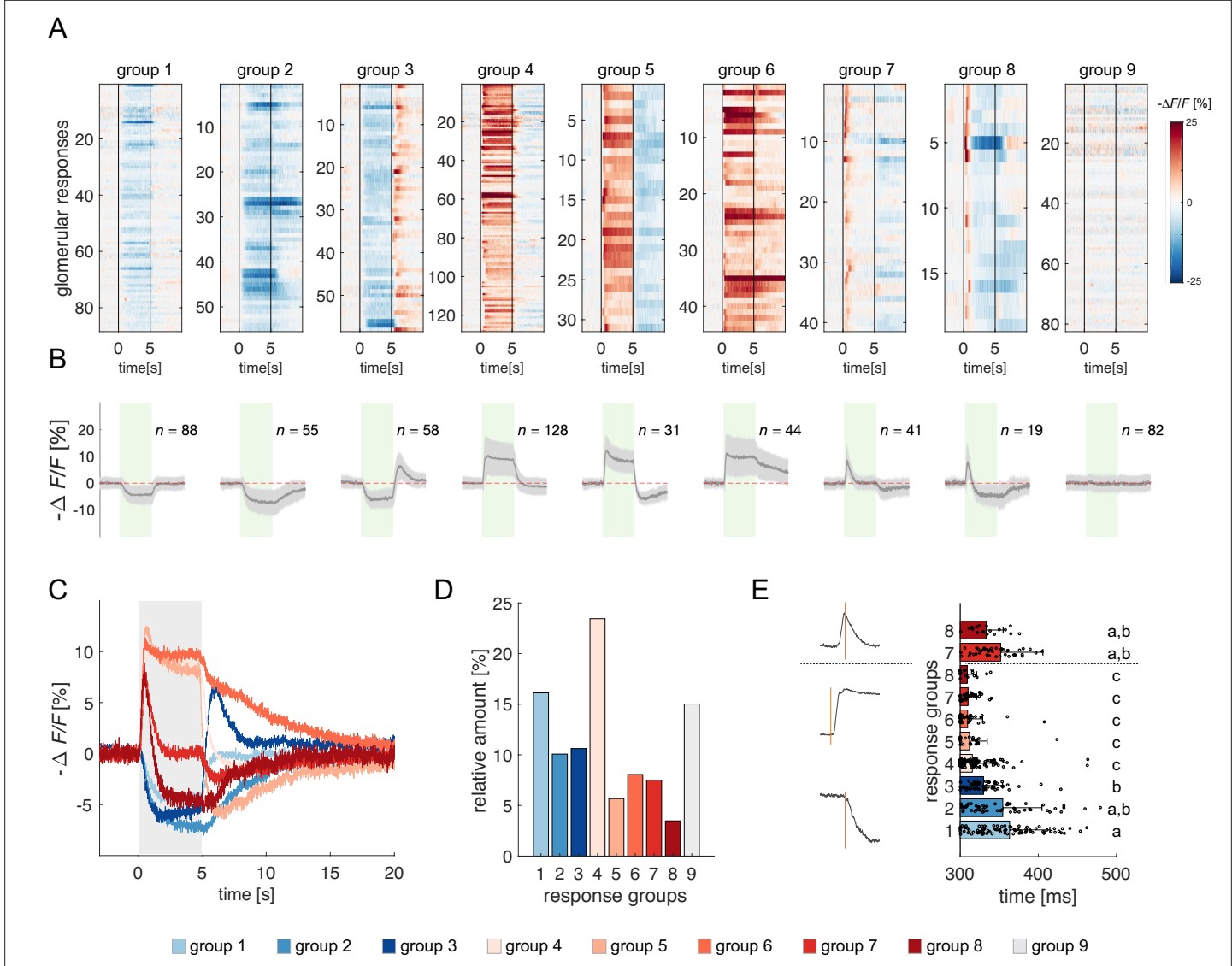

**Figure 2.** Clustering of projection neurons' response profiles. (**A**) All glomerular responses were clustered with supervision according to their activity (excitatory, inhibitory, non-responsive) during and after stimulus arrival. (**B**) Mean ± s.e.m. of all curves of the relative groups in (**A**). The green patch indicates the stimulus delivery interval. (**C**) Average curves of all response groups from (**B**) are superimposed. The grey patch indicates the stimulus delivery interval. (**D**) Relative amount of all response categories. (**E**) Latency of glomerular responses for excitatory and inhibitory profiles (groups 1–8). For groups 7 and 8, the latency of short excitatory response's termination was calculated. Letters refer to significative groups after Kruskal-Wallis statistical test and Tukey-Kramer correction. On the left side, exemplary traces of phasic and tonic excitatory responses and an inhibitory response are shown. The orange line indicates the latency of response onset or termination.

The online version of this article includes the following figure supplement(s) for figure 2:

**Figure supplement 1.** Odorant response profiles for each bee and stimulus.

stimulus-specific (*Honegger et al., 2011*; *Turner et al., 2008*). Hence, we enriched the model with a winner-takes-it-all feedback inhibition mechanism by forcing that only 10% of KCs receiving the largest summed input generate an action potential. Third, as in bees and locusts, MB have a 20 Hz oscillatory cycle (*Cassenaer and Laurent, 2007*; *Laurent and Naraghi, 1994*; *Popov and Szyszka, 2020*), we fed the model with experimentally recorded calcium signals resampled at a 20 Hz-sampling frequency. This approach allowed testing if a simple neural architecture – so far challenged with simulated datasets (*Ardin et al., 2016*; *Buehlmann et al., 2020*; *Peng and Chittka, 2017*) – could be used for understanding how real odour activity recorded at the level of PNs is transformed in the MB, and if the rules governing the model were sufficient to support appetitive olfactory conditioning.

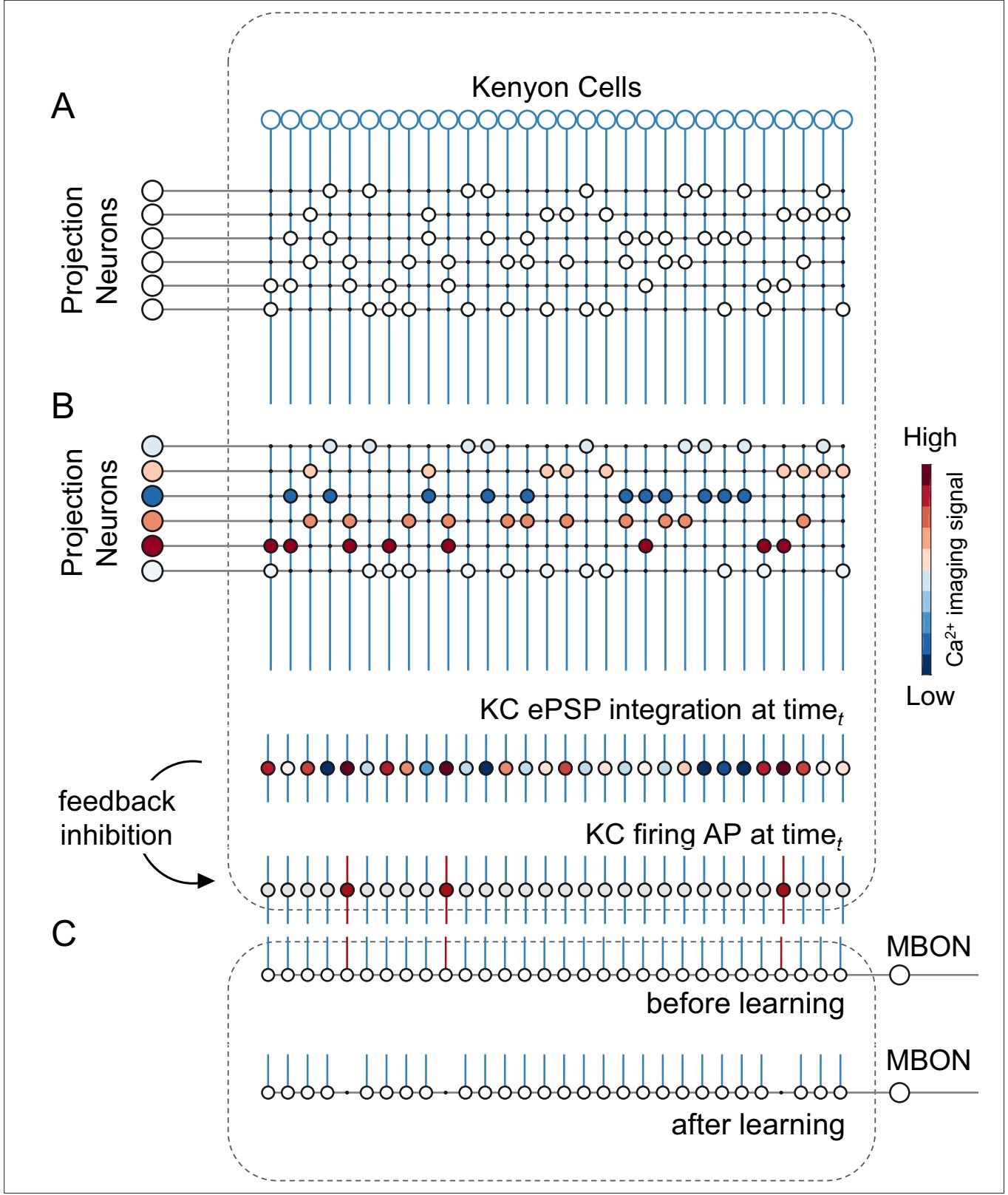

**Figure 3.** A simple neural network model for olfactory coding and learning. (**A**) The synaptic connectivity between AL PNs and KCs is built as a pseudorandom logic matrix where each column represents KC dendritic arborisation and each row PN axon terminals. Each KC receives synaptic contact – here represented as a white circle – from 30% of the PN population. (**B**) For each timepoint *t* of a calcium imaging recording, the measured activity level for each glomerulus (represented by a PN in the model) is projected onto all its synaptic connections with the KC population. Each KC,

*Figure 3 continued on next page*

*Figure 3 continued*

that is each column in the scheme, integrates all excitatory post-synaptic potentials. Finally, recurrent inhibitory feedback is simulated by imposing that only the 10% most active KCs will generate an action potential (AP) (red cells), while all others remain silent (grey cells). (**C**) A single appetitive MBON receiving input from all KCs was modelled. Based on a synaptic plasticity threshold parameter, all synapses from KCs to MBON that are activated with a frequency greater than allowed by the defined threshold during the learning window will be switched off. This phenomenon results in a decrease AP probability in the modelled MBON upon stimulation with the learned stimulus. Abbreviations: action potential, AP; antennal lobe, AL; excitatory post-synaptic potential, ePSP; Kenyon cells, KC; mushroom body, MB; mushroom body output neuron, MBON; projection neurons, PN.

## Model input and output: Simulating olfactory coding in the mushroom body

For each recorded individual AL ($n$=8), an MB simulation was generated and fed with the time series of the recorded glomerular activity obtained for that individual. Thus, for each odorant and individual, we simulated the time course of the firing of a virtual KC ensemble (*Figure 4A*). The analysis of KC population turnover (*Figure 4B*) showed that, before odorant onset, the population varies randomly between adjacent time points while it stabilises during olfactory stimulation, with a turnover rate of ~10%. The dynamics of KC recruitment across bees and odorants (*Figure 4C*) indicates that two odorant-specific sets of cells are recruited during an olfactory stimulation: the first one at stimulus onset, the second one at odour termination (see *Figure 4C*, $t$=5 s). Both populations are rather stable, with a turnover rate below the pre-stimulus baseline for up to 15 s after stimulus offset (turnover rate$_{\text{pre-stim}}$=0.61 ± 0.14; turnover rate$_{\text{ON\_0-5s}}$=0.17 ± 0.05; turnover rate$_{\text{OFF\_5-10s}}$=0.25 ± 0.04; turnover rate$_{\text{OFF\_10-15s}}$=0.33 ± 0.08). Correlation analysis of stimulus-induced responses (*Figure 5D and E*) showed a high correlation of neural activity across time points during olfactory stimulation and during the post-odour window both in the recorded PN population and in the simulated KCs. This indicates that the neural representation of an odorant is sufficiently stable to provide a neural image, which is coherent with itself during a 5 s stimulation window and during a~10 s post-stimulation window. Furthermore, the across-odorants correlation is rather high in the PN space (Pearson's correlation coefficient, $r$=0.4), but proximal to zero in the KC space, supporting the notion that the MB network increases stimulus discriminability, not only during but also after odour offset.

To further corroborate this idea, we compared the temporal relative trajectories obtained for the three odorants in the measured PN space and in the modelled KC space (*Figure 4F*). In both cases, response trajectories diverged at stimulus onset and returned to the centre of space after stimulus termination. However, the principal component analysis showed that odorants are better separated from each other and that they remained separated for a longer time in the KC space compared to the PN space. This indicates improved and prolonged stimulus discriminability in the MB with respect to the AL (*Figure 4—figure supplement 1*).

We next investigated the discriminatory power of the system across a larger set of odour response profiles. For this, we pooled together all response profiles acquired during the calcium imaging analysis (*Figure 1F*) and we artificially combined them to simulate 100 odorant response vectors, each comprising 30 different glomerular profiles (see Methods). Using these artificially combined odorant response profiles, we simulated the relative KC response time series and measured the correlation across all response vectors in the PNs (experimental data) and KCs spaces (modelled data *Figure 4G*). Notably, while response vectors calculated from PN data were highly correlated ($r$=0.71 ± 0.08), such a correlation significantly decreased in the modelled KC space ($r$=0.52 ± 0.10; Student's $t$-test between $r$ distributions of PN and KC data: $p$⬚ 0, $n$=4774), indicating that the signal transformation operated by the proposed MB neural architecture induces a strong decorrelation among odour signatures.

Overall, the proposed MB network can process experimentally acquired input data and produce physiologically plausible KC response patterns coherent with in vivo KC activity measurements (*Lüdke et al., 2018*). Moreover, it accounts for the expansion of the coding space from the AL to the MB (*Lin et al., 2014*; *Papadopoulou et al., 2011*; *Stopfer, 2014*), enhancing inter-stimuli decorrelation and confirming previous theoretical works based on artificial datasets (*Olshausen and Field, 2004*; *Peng and Chittka, 2017*). Importantly, these emergent properties of the model architecture are based on the known neuroanatomy and physiology of the insect brain and have been obtained without any optimisation aiming at reproducing specific features of olfactory coding.

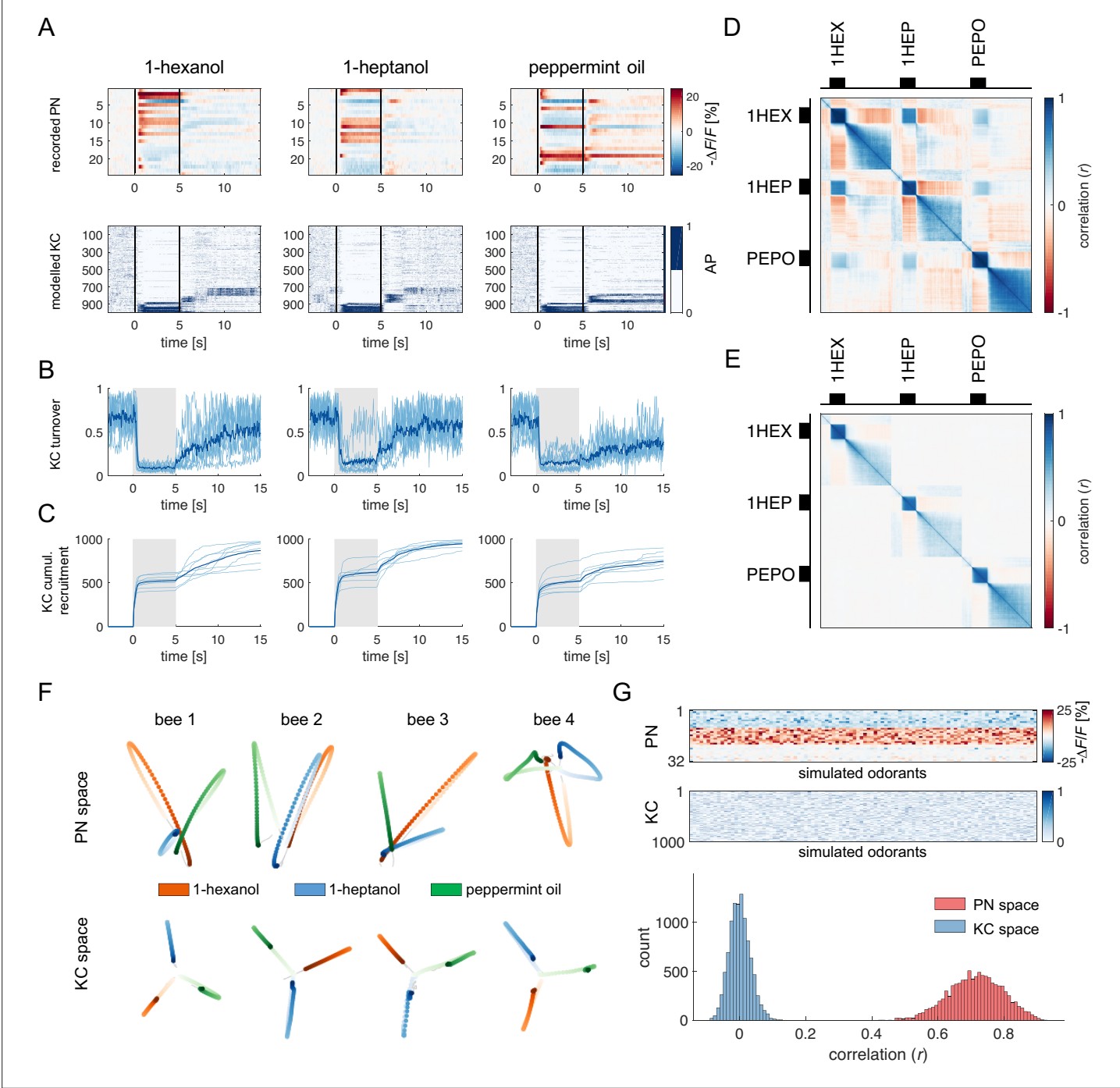

**Figure 4.** Olfactory representation in modelled Kenyon cells (KCs). (**A**) The first row displays the recorded PN responses of one honey bee to three odorants. The second row shows the transformation of the PN activity operated by the model to simulate the activity of 1000 KCs. In each plot, KCs are ordered for response strength during the onset and offset to better visualise activity clusters. (**B**) Turnover rate of recruited KCs across adjacent time points. Light blue traces refer to individual bees (mean of 10 MB simulations). Thick, dark blue traces indicate average curves across bees. (**C**) Cumulative recruitment of KC shows two main recruitment events at stimulus onset and offset. To observe odorant-related KC recruitment dynamics, the cumulative sum was initiated at stimulus onset. Light blue traces refer to individual bees (mean of 10 MB simulations). Thick, dark blue traces indicate average curves across bees. (**D, E**) Correlation matrix among time points of measured PN activity (**C**) and simulated KC activity (**E**). Mean responses to the three odorants were concatenated to allow observing within and across odorant correlations. (**F**) PCA of odorants' trajectories in the measured PN space (top) and in the modelled KC space (bottom) for four exemplary bees. Trajectories comprise a 10 s interval, ranging from odour onset (t=0 s, light) to 5 s after offset (t=10 s, dark). (**G**) The raster plot shows a set of 100 artificially combined glomerular responses (top), each simulating the neural representation of 100 similar odorants in the PN space. A second raster plot (middle) shows the representation of the same odorants in the modelled

*Figure 4 continued on next page*

*Figure 4 continued*

KC space. Note that odorant representation was considered as the mean activity of PN (or KC) during stimulus arrival (0.5–5 s). The histogram shows the distributions of between-odorant correlations computed among all pairs of odour vectors in the PN and KC space. A significant difference between the two distributions was tested with a Student's $t$ test ($p$ 0, $n$=4774).

The online version of this article includes the following figure supplement(s) for figure 4:

**Figure supplement 1.** Odorant-specific activity lasts longer in the Kenyon cells space.

**Figure supplement 2.** Dependence of Simulated MBON Activity on Synaptic Plasticity Threshold.

## The neural network model predicts appetitive behaviour

Appetitive classical conditioning relies on the coincidental activation of the neural elements representing the conditioned stimulus (CS) and the unconditioned stimulus (US). The latter is mediated by various neuromodulators, such as specific dopaminergic, as shown in the fly (*Burke et al., 2012*; *Liu et al., 2012*), or octopaminergic neurons, as demonstrated in the honey bee (*Hammer, 1993*).

The coincidental activation of CS and US neural elements induces plastic modulations in the MB output neurons (MBONs; *Hige et al., 2015*; *Okada et al., 2007*; *Owald et al., 2015*). Here, we assessed if the proposed MB model could reproduce empirical measurements of appetitive learning by introducing a time window, during which the weights of the recruited KC-to-MBON synapses could be downregulated from 1 to 0 (*Ardin et al., 2016*). This rule reflects MBON learning-induced plasticity, according to which they display a broadly tuned response, which is reduced in presence of a learned stimulus (*Amin and Lin, 2019*; *Aso et al., 2014b*; *Aso et al., 2014a*; *Cognigni et al., 2018*; *Cohn et al., 2015*; *Hige et al., 2015*; *Lyutova et al., 2019*; *Okada et al., 2007*; *Owald et al., 2015*). To implement such a rule, we introduce a synaptic plasticity threshold (*spt*) parameter, which defines the number of firing events of a given KC upon which its synaptic output weight is reduced from 1 to 0. In other words, it determines how active a synapse should be during the learning window before being switched off.

It is well known that the inter-stimulus interval (ISI) – that is the time elapsed from CS to US onsets – can influence the efficiency of Pavlovian learning (*Domjan, 2015*; *Holland, 1980*). The calcium signal dynamics and modelling analysis presented here (*Figure 4A–F*) suggest that such an ISI-dependent effect is the consequence of the strong temporal dynamics of the odorant neural representation. As the neural representation of the CS changes in time, a reward system recruited at different time points will interact with a different population of CS-recruited KCs. Because learning results from the coincidental activation of CS and US elements (*Pavlov, 1927*), an individual should respond to the different time points of an olfactory stimulation with different strengths, depending on the US arrival time experienced during conditioning. To test this hypothesis, learning-related MBON plasticity was modelled for four different CS/US temporal contingencies reflecting four conditioning paradigms, well-studied in honey bee (*Giurfa and Sandoz, 2012*; *Figure 5A*). Having a fixed CS stimulation interval from $t$=0–5 s, and a learning window (US) of 3 s, we simulated MBON learning with the following ISIs: *backward* pairing occurred when the US initiated at $t$ = –2 s (i.e. 2 s before CS onset), *early* pairing when it started at $t$=1 s (1 s after CS onset), *delay* pairing when it started at $t$=4 s (4 s after CS onset), and *trace* pairing when the US started at $t$=7 s (2 s after CS offset). First, we trained the model with the experimental PN response profile recorded for either 1-hexanol or peppermint oil (see *Figure 1*). One odorant was used as CS and the other as a novel odorant to test the specificity of the learning obtained under these experimental conditions. After learning, we modelled the MBON action potential (AP) probability upon the presentation of the CS (e.g., 1-hexanol) and the novel odorant (e.g., peppermint), and repeated this operation for the response profiles of all individuals. To account for the biological variability in the neural activities elicited by repetitions of the same stimulus, learning was assessed against the KC response profiles resulting from five CS and five novel odorant presentations, which were not included in the training dataset. As expected, we found that a trained MBON showed a larger decrease in firing probability when presented with the CS than with the unfamiliar stimulus, demonstrating learning (*Figure 5B*). Interestingly, the model predicts weak learning in the case of *backward* pairing – that is when US and CS are partially overlapping, but the US begins before the CS – whereas both *early* and *delay* conditioning predict strong learning, that is, a large CS-specific down-regulation of MBON firing probability (*Figure 5B*). Finally, in the case of *trace* conditioning, the model also produced a down-modulation of the MBON response. However,

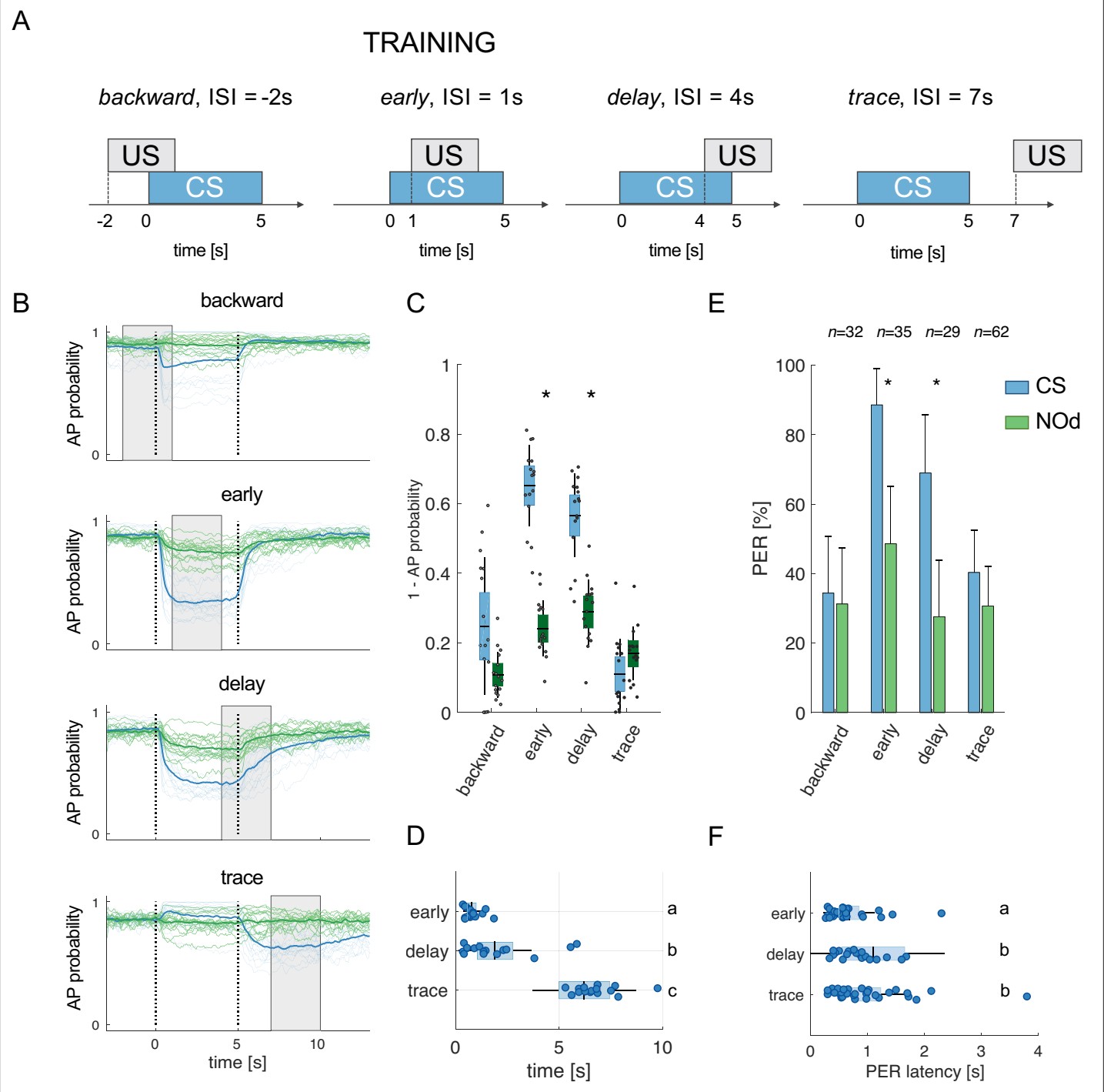

**Figure 5.** Modelled mushroom body output neurons can predict behavior. (**A**) Four experimental protocols differing for CS/US inter stimulus interval (ISI) were used for training the model (**B–D**) and for behavioural measurements of the proboscis extension reflex (PER) conditioning (**E, F**). Trained simulated MBONs, as well as conditioned honey bees were tested against the conditioned stimulus (CS, blue) and a novel odorant (NOd, green). (**B**) Time course of action potential (AP) probability of a trained MBON upon stimulation with the CS (blue) or with the novel odorant (green). Olfactory learning was modelled according to *backward*, *early*, *delay*, and *trace* conditioning protocols. Glomerular responses for eight bees to 1-hexanol and peppermint oil were used alternatively as CS and NOd. Data for responses to CS and NOd were pooled together ($n=16$ traces for each protocol). Thick trace: average AP probability profile; dotted vertical lines: stimulus onset/offset; grey bar: modelled learning window. (**C**) Distribution of the mean values of traces in (**B**) during CS and NOd stimulation. Because appetitive learning produces a decrease in MBON firing rate, the complementary value of the probability fraction provides a proxy for a learned appetitive response (Kruskal-Wallis statistical test: $p_{backward} = 0.0545$; $p_{early} = 1.42*10-6$; $p_{delay} = 1.04*10-5$; $p_{trace} = 0.0595$; $n=16$). (**D**) Latency of 90% of minimal MBON activity in response to the CS for early and delay conditioning protocols (Wilcoxon test early *vs*

*Figure 5 continued on next page*

*Figure 5 continued*

delay, p=0.0015; delay *vs* trace, *p*=0.0005. Letters indicate statistical groups according to the Wilcoxon test. Barlett's test for variance difference early *vs* delay, *p*=1.73*10$^{-6}$; delay *vs* trace, *p*=0.1808; early *vs* trace *p*=1.29*10$^{-8}$). (**E**) Memory retention test of honey bees 1 hr after absolute conditioning. Bars indicate the percentage of individuals showing proboscis extension reflex (PER) when presented with the conditioned (blue) or the novel odorant (green). Error bars indicate 95% confidence intervals. (Responses to the CS/NOd were compared with a McNemar test for binomial distribution; $p_{early}$ = 0.003, $p_{delay}$ = 0.004). (**F**) Latency of the PER to the conditioned stimulus at the 1 hr memory retention test (Kruskal-Wallis statistical test: early *vs* delay, *p*=0.036; early *vs* trace. *p*=0.041; delay *vs* trace, *p*=0.992. Letters indicate statistical groups according to the Kruskal-Wallis test. Barlett's test for variance difference across protocols: early *vs* delay, *p*=5.19*10$^{-6}$; early *vs* trace. *p*=0.020; delay *vs* trace, *p*=0.007; $n_{early}$ = 26; $n_{delay}$ = 20; $n_{trace}$ = 31).

this learnt response was shifted in time: it appeared only after the CS offset, that is, around the US expected arrival time (*Figure 5B*).

In addition, according to the model, high synaptic plasticity sensitivity (low *spt* values) facilitates non-specific learning, resulting in a strong generalisation effect. Conversely, MBON plasticity under high *spt* values requires a robust and stable KC activation. This scenario is not compatible with a learning interval that is only partially overlapping with the stimulation window (note that measurements are provided at 20 Hz, and in this context, a *spt* = 40 requires a synapse to be activated during two whole seconds – although not continuously – to be switched off) (*Figure 4—figure supplement 2*).

Stimulus representation is dynamic (see *Figures 1 and 2*). Hence, our model predicts that an MBON exposed to an *early* protocol should learn to identify the early components of the stimulus neural signature. Indeed, the model shows that the learned response (a down-regulation of MBON firing probability) occurs earlier in the *early* than in the *delay* contingency (Wilcoxon test for non-parametric paired data, *p*<0.001; *n*=16) and with lower response latency variability (Barlett's test *early* vs *delay*, *p*<0.011; *Figure 5D*). At the same time, it predicts that *trace* conditioning should rely on the identification of post-stimulus neural activity (*Figure 5B*). As such, the learned response occurs after odour offset and with a variability comparable to the one detected for the *delay* protocol (Barlett's test *delay* vs *trace*, *p*=0.1808; *Figure 5D*).

To test the model's predictions, we measured the timing of real behavioural responses (i.e. the proboscis extension) of bees trained under the same CS/US conditions and tested 1 hr later with the CS and a novel odorant. Because the model's predictions were generated based on the odour response maps elicited in the AL by 1-hexanol and peppermint oil, the same odorants were used for the behavioural experiments. In agreement with the model's predictions and previous reports (*Felsenberg et al., 2014*), *backward* conditioning led to scarce and unspecific responses (*Figure 5E*). In contrast, positive ISIs of 1 or 4 s (*early* and *delay* protocols) induced specific memory formation (McNemar test, $p_{backward}$ = n .s., $p_{early}$ = 0.003, $p_{late}$ = 0.004), with ~90 to 70% of the trained bees showing conditioned responses to the CS and ~40% showing a generalised response to the unfamiliar stimulus during the test. Video tracking of the proboscis extension response showed that bees receiving the reward 1 s after CS onset extended the proboscis earlier and with less variability than those that received the reward towards the end of CS delivery (Kruskal-Wallis test, p=0.036; Barlett's test for data variance comparison, p=5e10$^{-6}$; $n_{early}$ = 26, $n_{delay}$ = 20; *Figure 5F*). Finally, bees exposed to *trace* conditioning (ISI = 7 s) showed weak and aspecific learning - highlighting the complexity of the task (*Dylla et al., 2013*; *Ito et al., 2008*; *Paoli et al., 2023a*; *Szyszka et al., 2011*). However, in opposition to the model's prediction, the learners showed a PER latency comparable to the *delay* group (Kruskal-Wallis test, *p*=0.992) but with a broader distribution (Barlett's test for data variance comparison, *p*=0.007; $n_{delay}$ = 20, $n_{trace}$ = 31).

## Relative contribution of glomerular response dynamics to conditioned response latency

Early excitatory responses are confined to the initial part of the stimulus, while inhibitory responses – with a slower and delayed onset – show a delayed onset with respect to the excitatory responses. As such, they both contribute to the creation of an early (<1.5 s) and a delayed/stable (>1.5 s) odour signature (*Figure 2*). Such a temporal structure might provide the neural basis for the difference in response latency observed upon *early* and *delay* conditioning both in behavioural and modelled data. If so, the absence of such dynamics might increase odour response stability and prevent the differential response observed in MBONs subject to the two conditioning protocols.

To test this hypothesis, we used the glomerular response database (*Figure 1F*) to generate a set of simulated odorant responses, each comprising 32 glomerular response profiles selected among the different response groups (*Figure 2A and B*) based on the average occurrence of each group across all recorded odour response maps (see Methods section). As we did with real odorants (*Figure 5*), we used the simulated odorant responses to train modelled MBONs with *early* and *delay* protocols. Then, we assessed MBONs' firing rate upon stimulation with the conditioned and a novel stimulus. As with real odorant responses (*Figure 5*), also when using artificially generated odorant response profiles, MBONs showed an anticipated response when trained with the *early* protocol (*Figure 6A–C*, first row).

Next, we evaluated the relevance of the different glomerular response types in determining the different MBON response latency occurring upon training with the different CS/US contingencies. We selectively removed from the generated odorant response profiles all stable excitatory responses (groups 4–6; *Figure 6A–C*, second row), inhibitory responses (groups 1–3; *Figure 6A–C*, third row), short excitatory responses (groups 7 and 8; *Figure 6A–C*, fourth row), and both short excitatory and inhibitory responses together (groups 1–3, 7, 8; *Figure 6A–C*, fifth and last row). These modified versions of the simulated odorants were used to model MBON response dynamics upon *early* and *delay* conditioning. Interestingly, removing stable excitatory or inhibitory response profiles or short excitatory responses alone did not affect the difference in response timing between protocols. On the contrary, the removal of stable components such as long excitatory and inhibitory profiles seemed to enhance such difference in response latency (*Figure 6D*). Conversely, the combined removal of inhibitory and short excitatory responses from the odorant response patterns prevented the difference in MBON response latency between protocols.

In conclusion, reducing the richness in glomerular response dynamics by removing early excitatory responses and delayed inhibition increases the temporal stability of the neural representation of an olfactory stimulus and prevents the differential response latency observed between *early* and *delay* conditioning protocols.

## Discussion

Our results show that a simple but biologically realistic MB neural network model simulating essential aspects of MB connectivity can process real neurophysiological input and generate robust predictions that agree with empirical observations. Our model was fed with experimental PN responses acquired at high temporal resolution (*Bestea et al., 2022*; *Paoli and Haase, 2018b*). This approach provided a better description of the temporal dynamics of olfactory coding and a more realistic modelling of odour signal transformation and learning.

### Olfactory trajectories in the glomerular space are shaped by local inhibition

Because of the high temporal resolution of the functional imaging recordings, we could reliably measure odorant-induced latencies of excitatory and inhibitory PN responses. Excitatory responses occurred earlier than inhibitory ones and exhibited a smaller temporal variability. This suggests that excitation of PNs occurs in a single wave upon cholinergic input from the OSNs to the glomeruli of the antennal lobe, with a minor contribution – if any – of second-order excitatory local interneurons within the AL, which have been reported for *Drosophila* (*Chou et al., 2010*; *Olsen et al., 2007*) but so far not for honey bees (*Girardin et al., 2013*; *Krofczik et al., 2008*; *Schäfer and Bicker, 1986*). This finding also advocates that odour responses of OSNs may not display onset-latency variability as shown in the locusts (*Kim et al., 2023*). On the other hand, inhibitory PN responses and the termination of early excitatory PN responses occurred within the same time window, approx. 40 ms after the onset of excitatory responses, suggesting that lateral inhibition within the AL is responsible for triggering PN inhibition and terminating early excitatory responses (see *Figure 2*, groups 7 and 8). This is consistent with a significant presence of GABAergic local interneurons in the AL (*Schäfer and Bicker, 1986*), which reshape the olfactory message conveyed to higher order brain structures such as the MBs and the lateral horn (*Paoli and Galizia, 2021*). Notably, we observed that almost 50% of PN responses recorded contain an inhibitory component (see *Figure 2A*, groups 1–3, 5, 7, 8), underlining the dominant role of inhibitory AL interneurons in shaping odour trajectories in the glomerular space. These

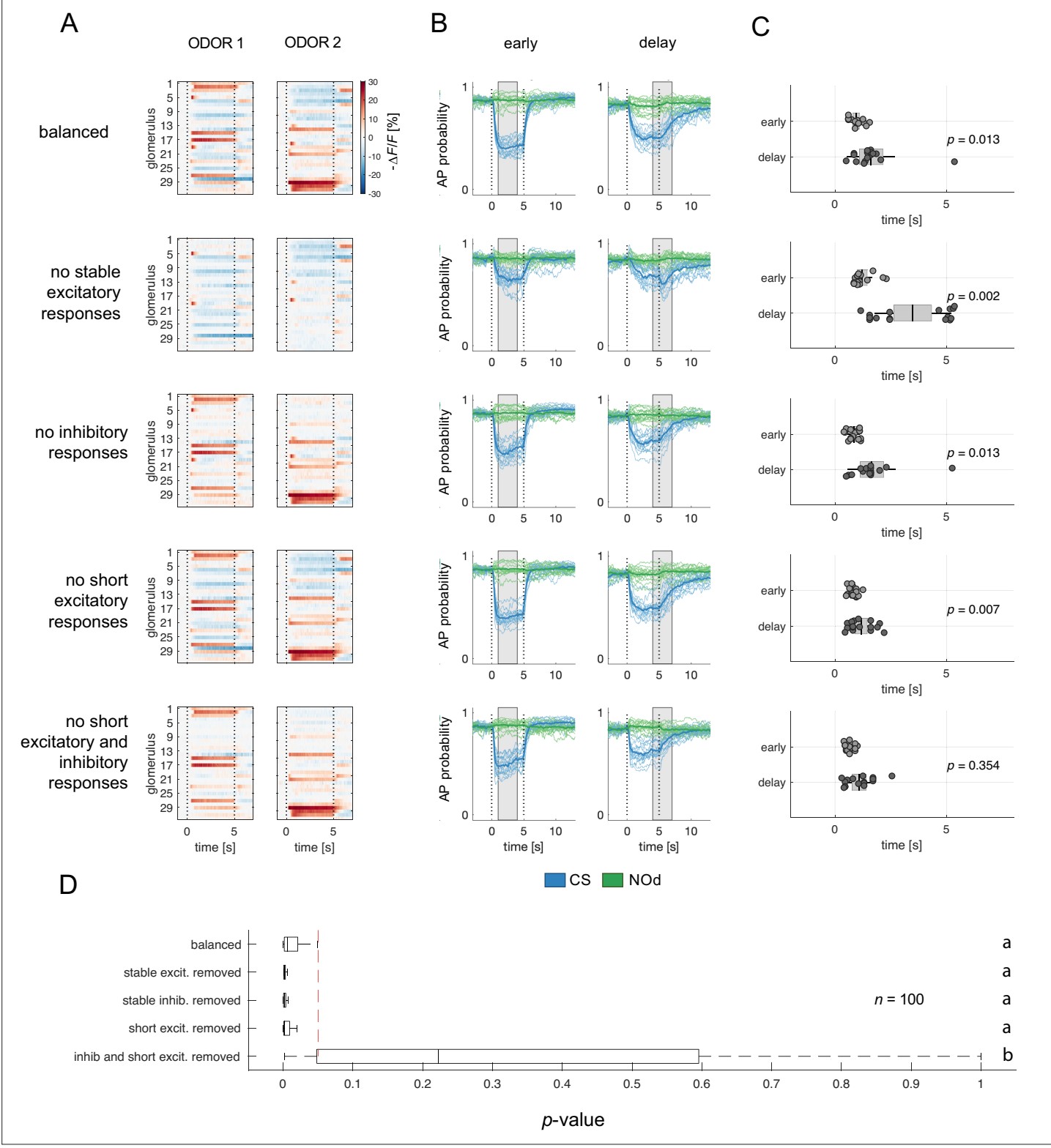

**Figure 6.** Relative contribution of glomerular response types to MBON response latency. (**A**) Two instances of odorant responses generated by artificially combining 32 glomerular response profiles from all response groups from *Figure 2* (first row). Response profiles were manipulated eliminating stable excitatory responses (second row), inhibitory responses (third row), short excitatory responses (fourth row), and short excitatory and inhibitory responses together (last row). Dotted lines indicate stimulus onset and offset. (**B**) Time course of action potential (AP) probability of a trained MBON upon stimulation with the conditioned stimulus (CS, blue) or with the novel odorant (NOd, green). Olfactory learning was modelled according to early

*Figure 6 continued on next page*

*Figure 6 continued*

and delay conditioning protocols and with the balanced (first row) and modified simulated odorants (second to last row). Conditioned stimulus and novel odorant are in blue and green, respectively (*n*=16). Thick trace: average AP probability profile; dotted vertical lines: stimulus onset/offset; grey bar: modelled learning window. (**C**) Latency of 90% of minimal MBON activity in response to the CS for *early* and *delay* conditioning protocols (Wilcoxon signed rank test early *vs* delay; *n*=16). (**D**) Distribution of Wilcoxon signed rank test *p*-values of 100 simulations of (**C**). The red dashed line indicates a significance level of 0.05; letters indicate significance groups.

observations are in contrast with previous calcium imaging analyses, where only 2–10% of responses had an inhibitory nature (*Krofczik et al., 2008*; *Szyszka et al., 2005*), but coherent with electrophysiological recordings showing a rich response diversity and prominence of inhibitory activity (*Wilson et al., 2004*). Differently from *Drosophila*, where both local inhibitory and excitatory interneurons contribute to the olfactory tuning (*Chou et al., 2010*; *Huang et al., 2010*; *Shang et al., 2007*; *Yaksi and Wilson, 2010*), our analysis indicates that, in the honey bee AL, glomerular responses are modulated mainly by a local inhibitory network. This conclusion is supported by a previous study showing that pharmacological activation of a single glomerulus by topical acetylcholine application provokes inhibitory but not excitatory responses in other glomeruli (*Girardin et al., 2013*). Indeed, while the presence of excitatory local interneurons has been predicted (*Malaka et al., 1995*), such prediction has never been supported by experimental evidence. A population analysis suggests that the varied response profiles are equally distributed across individuals and can be triggered by different olfactory stimuli. Still, it remains to be understood to what extent the response profiles for each odour/glomerulus combination are conserved across individuals.

Overall, these findings highlight possible differences between the ALs of bees and fruit flies regarding the nature of the local interneuron populations modulating glomerular activity. Thus, they cast caution in prioritising a single species as a representative general model of insect olfaction.

## Smell and aftersmell: An odorant-specific afterimage

Almost 50% of all PN responses in our database displayed either a prolonged post-stimulus activity (see *Figure 2*, groups 2 and 6) or a change in response direction after stimulus termination (groups 3, 5, 7, 8). This results in the recruitment of an offset-specific population of KC – both in our model and in the literature (*Ito et al., 2008*; *Lüdke et al., 2018*) – that creates a neural image of the post-odour in the MB. Thus, the post-odour representation is different from the neural image of the odorant itself and provides a stable, slowly decaying after-image, which is stimulus-specific and temporally associated with the neural correlate of the odorant itself. This is in agreement with the analysis conducted by *Patterson et al., 2013* on the olfactory representation in M/T cells in mice, which shows that the post-odorant activity is rather conserved and stimulus-specific even across concentrations and for about a 10 s window. Our modelling analysis predicts that the after-stimulus signature of an odorant is more distinct and stable across time in the MB than in the AL (*Figure 4—figure supplement 1*). This could occur because the activity across the KC population is normalised by the MB feedback neurons operating a gain control, which stabilises odorant-responses signature in the MB – including the post-odorant activity – while the input signal in the PN is fading.

The existence of a post-odour signature has important implications for understanding the neurobiological basis of trace learning (*Dylla et al., 2013*; *Paoli et al., 2023a*), a conditioning protocol in which a stimulus-free time window separates CS and US. Previous studies reported that an odorant cannot be predicted by its post-odour signature (*Lüdke et al., 2018*; *Szyszka et al., 2011*). Nevertheless, as shown in previous studies (*Galili et al., 2011*; *Lüdke et al., 2018*; *Patterson et al., 2013*; *Szyszka et al., 2011*), the neural activity associated with the OFF response is stimulus-specific and reproducible, suggesting that the post-odour could be physiologically similar to the onset of a second odorant (in the same way as the back-taste of wine could remind us of an unrelated ingredient). If so, the odour/post-odour pattern could be processed as an individual neural object, effectively extending the duration of the nominal, conditioned odour. Alternatively, it is also possible that the stimulus specificity of the after-smell signature and its temporal contiguity with the CS may enable some form of second-order conditioning. Certainly, the behavioural analysis within this study indicates that honey bees subjected to trace conditioning display the conditioned reflex already during odour arrival. This indicates that bees do not form simply the association between the after-smell (as CS) and the sugar reward (US). Conversely, it supports the idea that trace learning requires some form of prolongation

of the neural trace of the CS until US arrival (*Dylla et al., 2013*; *Paoli et al., 2023a*). Interestingly, the duration of the post-odour neural signature reported by us and by previous studies (*Galili et al., 2011*; *Patterson et al., 2013*; *Szyszka et al., 2011*) is about 10–15 s. As such, it is compatible with the duration of the CS/US gap in trace learning experiments (*Galili et al., 2011*; *Paoli et al., 2023a*; *Szyszka et al., 2011*; *Wystrach et al., 2020*), suggesting the possibility of its implication in this form of learning.

## Mushroom body modelling predicts associative learning scores and response latency for different conditioning protocols

To investigate if the proposed neural network model explains the temporal features of associative learning, we simulated CS delivery by feeding the model with experimental PN calcium imaging time series and applied a learning rule based on the downregulation of recurrently recruited synapses in the MB. A learning time window of 3 s was defined according to four well-studied protocols for associative conditioning, which correspond to four different ISI situations (–2 s,+1 s,+4 and+7 s) (*Bitterman et al., 1983*; *Giurfa and Sandoz, 2012*). A simulation of the action potential probability of an MBON after olfactory conditioning revealed that differentiation between the CS and a novel odorant was observable whenever the CS anticipated the US (ISIs of +1 s and +4 s) but less successful in the *backward* (ISI of –2 s) and *trace* protocols (ISI of 7 s). Generalisation errors were also observed in a similar way as in behavioural experiments. Notably, in this study, we adopted a backward contingency with a 1 s overlap between the US and the CS (*Figure 5A*). This differs from the more traditional configuration, where the CS is delivered after US termination, that is, without any temporal overlap. In this case, the model would predict no learning at all, as observed in honey bee behavioural experiments (*Felsenberg et al., 2014*; *Hellstern et al., 1998*). Hence, we tested the model's prediction in the form of *backward* conditioning where US and CS are partially overlapping. In this case, the model predicts a weak appetitive association (see *Figure 5B, C and E*). This result is coherent with a previous behavioural report, where bees subject to a similar protocol showed a weak response to the conditioned odorant (*Felsenberg et al., 2014*). In the *early* and *delay* configurations, a CS-specific associative memory was formed, both in the model (*Figure 5C*) and in the behavioural tests (*Figure 5E*). However, the model predicts a difference in response latency and latency variability depending on relative CS/US arrival (ISI of +1 or+4 s): a learning window placed closer to stimulus onset (*early* protocol) resulted in early and temporally precise MBON responses, whereas a delayed learning window produced a similar learning score, but with higher response latencies and temporal uncertainty.

Experimental results based on the same odorants that were fed to the model confirmed the model's predictions. They showed that *early* and *delay* protocols yielded comparable results in terms of learning score and CS-specificity while differing in terms of CS latency and precision (*Figure 5C–F*). These observations support the idea that associative learning occurs upon coincident detection of the CS and US neural elements. In fact, because the CS neural representation changes dynamically during the olfactory stimulation, a positive association with the initial odour signature will trigger an earlier and temporally precise conditioned response. Conversely, the association with the later part of the olfactory signature will elicit a delayed response with greater temporal variability.

Finally, both the model and data (*Figure 5C and E*) show that *trace* conditioning yields lower and less specific learning scores. However, while the model predicts that learners should respond around the expected US arrival time (i.e., during the after-smell component, *Figure 5D*), empirical data show that the individuals who learned the association extend their proboscis during (and not after) the olfactory stimulation. Such a discrepancy in measured and predicted response latencies suggests that trace conditioning cannot be explained by the simple temporal coincidence of a reward system's activation and a pool of KC recruited at odour offset. In order to trigger a conditioned response anticipating the expected reward time, the network must have learned the nominal CS stimulus and not simply its after-smell. Thus, a higher order cognitive process, possibly involving attention mediated by the serotonergic system, could contribute in prolonging the memory trace of the conditioned stimulus (*Paoli et al., 2023a*; *Zeng et al., 2023*).

*Vrontou et al., 2021* showed that activation of the fruit fly dopaminergic system at the end of a stimulus or after odour offset can, in certain cases, regulate the MBON spiking rate already at odour onset (*Vrontou et al., 2021*). This suggests that delayed reward system activation can modulate an olfactory stimulus's early neural coding. Consequently, the CS/US temporal contingencies used in

studies on *delay* and *trace* protocols may trigger an earlier-than-expected behavioural response by inducing learning-related neural plasticity in the early components of a CS response.

## Encoding time without encoding time

*Pavlov, 1927* proposed that the timing of a conditioned response could be adjusted to coincide with the expected arrival of the unconditioned stimulus (*Pavlov, 1927*). In his view, this capability involved two cognitive processes: learning to suppress the responses to the earlier phases of the conditioned stimulus and – at the same time – learning to delay the conditioned response until the US onset. Drew et al. showed that in goldfish, the timing of the conditioned response is influenced by the CS/US inter-stimulus interval, and its precision improves across learning trials (*Drew et al., 2005*). Importantly, this improvement was due to increased accurately timed responses rather than decreased off-time ones. Unlike in Pavlov's hypothesis, these findings suggested that timing acquisition during associative conditioning relies on learning to respond at the correct moment rather than to avoid responding at the wrong time. Experimental evidence in mammals indicates that their ability to adjust the latency of the conditioned response based on the US/CS relative onset depends on the cerebellar neural network (*Perrett et al., 1993*; *Garcia-Garcia et al., 2024*). Targeted lesions to the cerebellar cortex significantly impair this ability, suggesting that the adaptive timing of conditioned responses relies on a different mechanism than the one supporting the association. Although the neural mechanisms governing timing in the cerebellum are not yet fully understood, the involvement of this brain area in sensory-motor timing is well-established (*Kirkpatrick and Balsam, 2016*; *Paton and Buonomano, 2018*).

While the influence of the inter-stimulus interval on learning is well documented also in invertebrates (*Giurfa and Malun, 2004*; *Szyszka et al., 2011*; *Vogt et al., 2015*), the ability of relatively small insect brains to time the conditioned response based on the expected reward arrival remains unclear. Previous studies have shown that the duration of the inter-stimulus interval experienced during learning can influence the conditioned response's latency in bumble bees (*Boisvert and Sherry, 2006*) and honey bees (*Szyszka et al., 2011*). Both studies indicate that bees trained with a longer ISI exhibited a larger latency in their response to the conditioned stimulus. This provided clear evidence that CS-related information can modulate response timing. However, the appetitive response largely anticipated the expected reward time, and a clear interpretation of time tracking in insect brains is still missing.

Our model shows that this time contingency can be naturally encoded simply because the olfactory representation in the KCs evolves through time in a deterministic fashion. Thus, what is learnt during the US presentation is the KC configuration of the CS at this particular time. In other words, the MBON learns which subpopulation of KC is active during reward arrival rather than the expected reward time. Interestingly, because the KC representation of the CS does not evolve linearly with time (*Figure 4A–C*), the CS/US time contingency is not encoded perfectly. Remarkably, the model captures the experimental measurements of the proboscis response latency in bees, which indeed tend to respond earlier than the expected reward (*Figure 5D and F*). However, such reasoning may not apply to trace conditioning protocols – when a temporal gap separates CS offset and US onset. In this case, higher order neural processes that are not captured by our model may contribute to extending the stimulus-related information required for associative learning (*Paoli et al., 2023a*).

As often in insect research, heuristics can provide simple solutions to apparently complex cognitive problems. In this case, a mechanism based on the temporal dynamics of the conditioned stimulus, explains the bees' systematic errors at encoding proper CS/US contingency. Whether, in addition to this process, other mechanisms enable bees to track time contingencies remains to be seen.

## Conclusive remarks

In this study, we adopted a fast calcium imaging approach to investigate the dynamical representation of odorants in the antennal lobe. Our findings revealed that about 50% of glomerular responses had an inhibitory component and that local inhibition played a crucial role in shaping the initial PN activity and defining odour trajectories in the glomerular space. We observed that half of the PN response profiles prolonged their activity or changed response direction after stimulus termination, generating a stable, odorant-specific after-smell. Although the after-smell might be very different from the odour

signature, it is specific and temporally contiguous, thus providing the means to prolong the neural representation of the stimulus itself.

To investigate the logic of olfactory transformation and learning in the mushroom body, we constructed an MB neural network model based on the known connectivity in the insect brain. We showed that the model can process experimentally acquired input data and produce realistic KC response patterns. Our analysis suggests that the mushroom bodies enhance olfactory discrimination ability and stabilise the olfactory after-image, effectively prolonging the stimulus signature after odour offset.

Finally, we implemented a learning rule to modulate MB output neurons firing rate based on associative learning. We verified the model's predictions with behavioural measurements on real honey bees subjected to different learning protocols. The coherence between measured and predicted behavioural scores suggested that the rules behind this neural model – that is random connectivity, feedback inhibition, and MBON plasticity – are sufficient to generate relevant predictions for associative learning upon different learning protocols. In addition, by testing the model against physiological recordings of different stimulations, we showed that the model's predictions are robust to noise and biological variability.

Because the model incorporates not only neural assumptions pertaining to the honey bee olfactory system but also to the fruit fly and locust olfactory systems, similar principles may guide odour signal transformation at the level of the MB in appetitive associative learning across insects. This generality does not apply, however, to the nature of odour reshaping by local interneurons observed in the AL of bees, which seems to differ from that occurring in flies, thus highlighting the fact that convergence, but also differences, can characterise the functional architectures of olfactory systems in different insect species.

## Methods

### Experimental model

Experiments were performed on honey bees *Apis mellifera* reared in outdoor hives at the experimental apiary of the Research Centre on Animal Cognition (CNRS, Toulouse, France) situated in the campus of the University Paul Sabatier. In all cases, honey bee foragers (>3 week old) were used. No institutional permission is required for experimental research on honey bees.

### Projection neurons labelling

Honey bee foragers were collected the day before the experiment at an artificial feeder, to which they were previously trained, and projection neurons were labelled for calcium imaging analysis as previously described (*Paoli et al., 2017*; *Sachse and Galizia, 2002*). Shortly, bees were fixed in a 3D-printed holder with soft dental wax, antennae were blocked frontally with a drop of eicosane (Sigma Aldrich, CAS: 112-95-8). A small window was opened in the head cuticle, and glands and tracheas were displaced to expose the injection site. The tip of a borosilicate glass needle coated with Fura-2-dextran (Thermo Fisher Scientific Inc) was inserted between the medial and lateral mushroom body calyces, where medial (m-ACTs) and lateral antenno-cerebral tracts (l-ACTs) cross. After dye injection, the head capsule was closed to prevent brain desiccation, and bees were fed ad libitum with a 50% sucrose/water solution. On the following day, antennal lobes were exposed to allow optical access, and the brain was covered in transparent two-component silicon (Kwik-Sil, WPI). Although the injection procedure may result in variable levels of PN labelling, the reproducibility of response amplitudes and the lack of labelling bias for specific glomeruli indicate that the loading procedure is reproducible.

### Calcium imaging analysis and signal processing

Undiluted solutions of 1-hexanol (Sigma-Aldrich, CAS:111-27-3, 7.98 M), 1-heptanol (CAS:111-70-6, 7.05 M) and peppermint oil (CAS: 8006-90-4, 898 g/L) were delivered to the bees using an Arduino Uno-controlled automated olfactometer (*Bestea et al., 2022*; *Raiser et al., 2017*). Briefly, 1 mL of pure odorant was placed in a 20 mL glass vial. During an olfactory stimulation, the headspace concentration of volatile odorants is injected into a clean airflow, where it is diluted approximately 10 times before reaching the honey bee antennae. The odorous flow exiting the olfactometer is weak but

detectable by a human nose, in the range of honey bee olfactory sensitivity (*Carcaud et al., 2015*; *Gil-Guevara et al., 2022*; *Wright and Smith, 2004*). Odorants were alternated and presented 20 times on a 5/25 s ON/OFF configuration. Calcium imaging recordings were conducted with a straight Leica SP8 scanning microscope (Leica Microsystems, Germany) equipped with a SpectraPhysics InSight X3 multiphoton laser tuned at 780 nm for Fura-2 excitation. All images were acquired with a water immersion 16 x objective (Leica HC FLUOTAR 16 x/0.6 IMM CORR, Leica Microsystems, Germany), at 64x64 pixel resolution and ~127 Hz.

Calcium imaging data were analysed with custom-made MATLAB (MathWorks Inc) scripts. The baseline signal, calculated as the mean fluorescence during the one second before stimulus onset, was used to calculate baseline-subtracted and normalised stimulus-induced glomerular activity (ΔF/F). The normalised activity was multiplied by –1 to display excitatory/inhibitory responses as positive/negative relative changes (-ΔF/F). Time series of glomerular activity were averaged across stimulus repetitions for response profile clustering analysis. Glomeruli were hand-selected based on a morphological and functional map. To facilitate glomerular identification, the odour response maps for the three stimuli were merged into an RGB image: 1-hexanol was set as the red channel, 1-heptanol as the green, and peppermint as the blue. Thus, glomeruli excited only by peppermint appear in blue, glomeruli excited only by 1-hexanol appear in red, and glomeruli excited by a combination of the three channels will result in different colour shades. Glomerular response profiles were calculated as the mean intensity of an area of 5-by-5 pixels around the centre of the glomerulus across time. Stimulus representation stability (*Figure 1E*) was assessed with a Person's correlation analysis between vectors of the mean glomerular response of different trials. Glomerular response vectors' stability was calculated before, during and after olfactory stimulation.

## Mushroom body modelling

We used MATLAB R2018a to investigate how the recorded PN activity would affect KC responses of a MB network model.

### Projection neurons to KC connectivity

Real projection neurons' activity acquired via calcium imaging analysis provided the input to a simple model of the mushroom body neural circuits. For each experimentally measured honey bee, approximately 26 glomeruli were detected. Considering that each glomerulus hosts multiple PNs, for each glomerular signal, we modelled 3 PNs, each with its own independent connections with a subset of the 1000 modelled KCs. Connections between PNs and KCs were modelled as pseudo-random, with each KC receiving connections 30% of the input PN population, as observed in bees (*Szyszka et al., 2005*) and flies (*Litwin-Kumar et al., 2017*; *Caron et al., 2013*). PN-to-KC connections were assumed to be binary: either 0 (not connected), or 1 (connected), and the connectivity scheme is stable in a given MB model and not subject to plastic changes. This was implemented as a logic connectivity matrix (*W_PN_to_KC*) of size *number_of_PNs ×number_of_KCs*. Experimental PNs activity (*PNactivity*) was resampled at 20 Hz for modelling analysis (50ms temporal resolution) to match the 20 Hz oscillatory cycle detected in the MB (*Cassenaer and Laurent, 2007*; *Laurent and Naraghi, 1994*; *Popov and Szyszka, 2020*). For every time step *t*, the integrated input to each KCs (*KCepsp*) was calculated as the matrix product of the input neuron activity vector (*PNactivity*) and the logical connectivity matrix. Notably, we considered that a glomerulus is completely silent only at minimal inhibition. For this reason, at any time point, also glomeruli with negative ΔF/F values contribute to determining a KC integrated input. This integrated input of each KC is analogous to its excitatory post-synaptic potential (EPSP).

KCepsp(t) = PNactivity(t) × W_PN_to_KC.

### Winner-takes-it-all inhibitory feedback element

Whether a KC will fire an action potential depends not only on the sum of its input signals but also on the inhibitory activity of the MB feedback neurons (*Rybak and Menzel, 1993*; *Zwaka et al., 2018*). These neurons act on the whole KC population, providing that the KCs receiving the highest summed input will produce an action potential (AP). To account for such effect, and based on known MB neurophysiology (*Honegger et al., 2011*; *Peng and Chittka, 2017*; *Turner et al., 2008*), we imposed

that only the 10% most active *KCepsp* (i.e., receiving the strongest summed input) will fire an action potential.

If KC(*i*)epsp(*t*) > 10h percentil of KCepsp(*t*), then KC(*i*)activity(*t*) = 1; else KC(*i*)activity(*t*) = 0.

This outputs a binary vector of firing pattern across the whole KC population (*KCactivity*), where ones correspond to action potential and zeroes to silent KC. This operation is repeated for all epochs to obtain the time course of the activity of the KC population before, during and after odour arrival. To ensure the robustness of the model and account for the variability of the initial parameters, the MB simulation was run 10 times for each experimental AL input dataset. The results were then averaged over the 10 MB network simulations. An exemplary data subset and the main MATLAB scripts used to model KC activity and MBON learning have been uploaded to GitHub and are freely accessible at GitHub (copy archived at *Paoli, 2024*).

## KC-to-MBON model and learning-induced plasticity

Evidence in *Drosophila* and in the honey bee suggests that the default state of an appetitive MBON – that is, an MBON departing from an MB lobe innervated by a dopaminergic neuron conveying reward information – is to be broadly responsive to neutral stimuli (*Okada et al., 2007*; *Owald et al., 2015*). Under the assumption that similar principles are highly conserved among insects, we modelled an MBON receiving connections from all KCs and with all connections having an initial synaptic weight = 1 to reflect a generalised odour tuning. This provides a broadly-tuned MBON that responds equally to any pattern of KC activity. Upon learning, MBON synapses that are repeatedly recruited are switched off (=0). As a consequence, a trained MBON retains a high AP probability in the presence of an unfamiliar stimulus but a lower one in response to the conditioned odorant (or to an odorant that is represented by a similar KC population). Two parameters guide learning: a synaptic plasticity threshold (*spt*) and a learning window. The variable *spt* defines the number of firing events needed for a given KC to induce its output synaptic depression (i.e., KC(*i*)_to_MBON synaptic weight switch from 1 to 0). For the learning tests, a *spt* value of 15 was adopted. This value falls in the middle of the range tested in our simulations (from 1 to 40). It is stringent enough to hinder plastic modulation of rarely recruited KCs, while still requiring repetitive activation (i.e., a synapse must be active at least 750ms during a 3 s learning window) before allowing learning-induced plasticity.

The learning window provides a temporal restriction of such a learning rule, analogous to the reward/US delivery window in an appetitive conditioning protocol. Different CS/US contingencies were tested. Inter-stimulus intervals (ISI) of –2, 1 and 4 were used, where such values indicate the onset of a 3 s US window with respect to the onset of a 5 s CS window. Throughout the manuscript, we referred to the three learning protocols as *backward* (ISI = –2), *early* (ISI = 1), and *delay* (ISI = 4). The training was modelled using the mean glomerular responses to the first five stimulations of 1-hexanol or peppermint oil; memory retention was tested against the next 5 stimulations of both odorants (one acting as the conditioned stimulus, the other one as the novel odorant).

### Odorant response profiles simulation

The response profile database (*Figure 1F*) was repeatedly subsampled to generate 100 odorant response maps, each comprising 30 glomerular responses. The proportion of response types (inhibitory, excitatory, non-responsive, *etc.*) was maintained constant and equal to the mean proportions across the entire population to generate a batch of similar glomerular response maps (representing perceptually similar odorants *Guerrieri et al., 2005*; *Figure 4G*). Projection neurons (or KC) response vectors used for the Pearson correlation analysis are constituted by the average response for each PN (or KC) during olfactory stimulation (*t*=1–4 s). For *Figure 6*, 16 odour response profiles were generated by assembling 32 glomerular response profiles from the database. Each simulated odorant contained the same number of glomerular profiles for each type: 5 inhibitory responses (group 1); 3 long inhibitory responses (group 2); 3 inhibitory-then-excitatory (group 3); 7 excitatory (group 4); 1 excitatory-then-inhibitory (group 5); 2 long excitatory (group 6); 2 short excitatory (group 7); 1 short excitatory then inhibited (group 8); 4 unclassified (group 9); 4 unresponsive profiles.

### Proboscis extension response latency

Bees were collected in the afternoon at the institute's outdoor beehives, kept in custom-printed plastic cages in groups of 16 individuals, and provided with 240 μl of 50% sugar/water solution (an average

of 15 µl for each individual). On the next morning, they were harnessed in plastic tubes, blocked in place with tape, and fed 3 µl of sugar solution. Three hours later, they were exposed to an absolute olfactory conditioning protocol (*Villar et al., 2020*). During conditioning, bees were placed in front of the odour delivery device (the same one used for the calcium imaging analysis) and exposed to clean air for 15 s (familiarisation phase), to the odorant for 5 s, and to clean air again for another 20 s. Three different intervals were selected for delivering the sugar reward: either from 2 s before odour onset to 1 s after (*backward conditioning*), from 1 to 4 s from odour onset (*early conditioning*), from 4 to 7 s from odour onset (*delay conditioning*), or from 7 to 10 s from odour onset (*trace conditioning*). Bees were exposed to four rewarded trials, with a 10 min inter-trial-interval. Memory was tested 1 hr after the last conditioning trial. During the memory test, videos were acquired with a commercial video camera at 30 fps. A LED connected to the odour delivery device and in synch with the olfactory stimulation was used as a marker for the starting timepoint for the measurement of proboscis extension latency. Peppermint oil and 1-hexanol were used as olfactory stimuli. They were used non-diluted and their role as conditioned stimulus (CS) or novel odorant (NOd) was balanced between bees. During the test, half of the bees were exposed first to the CS and then to the NOd, while the remaining half was tested in the opposite order. A 50% sugar/water solution (w/w) was used as unconditioned stimulus (US).

## Statistical analysis

For MBON learning simulations, a total of 16 MB networks were generated and trained with the PN response profiles to 1-hexanol ($n=8$) or peppermint oil ($n=8$) (*Figure 5*), or with computationally assembled odorant profiles (*Figure 6*). After training, each MBON was tested against the response profiles of the CS and of the novel odorant to assess AP firing probability to the learned and the unfamiliar stimulus. For each simulated MBON, five memory tests against the five repetitions of the CS and of the NOd were performed. To quantify stimulus response to the conditioned/novel stimulus, the mean firing probability during the time-window of stimulus arrival was calculated. The difference between the responses to the CS and to the NOd was tested with a Kruskal-Wallis statistical test (*Figure 5C*). The latency of MBON response to the CS was calculated as the latency to reach the 90% of minimal firing probability upon stimulation. The difference in latency between protocols was tested with a Wilcoxon signed rank test (*Figures 5D and 6C*); the difference in latency variance was assessed with a Barlett's test (*Figure 5D*).

Proboscis extension response was used to assess memory retention 1 hr after the last absolute conditioning trial. Conditioned stimulus specificity was tested with a McNemar test. The latency of CS-specific PER was measured as the first frame after stimulus onset, where the proboscis trespasses the imaginary line between the open mandibles (*Figure 4E*). Differences between response latencies after early and delay conditioning were assessed with a Kruskal-Wallis test as well as with Barlett's test for difference in variance among datasets (*Figure 4F*).

## Acknowledgements

This work was supported by the MSCA long-term fellowship to MP, the ERC Starting Grant ('Emerg-Ant' 759817) to AW, and the ERC Advanced Grant ('Cognibrains') to MG, who also thanks the Institut Universitaire de France (IUF), the CNRS, and Sorbonne University for generous support.

## Additional information

### Funding

| Funder | Grant reference number | Author |
|---|---|---|
| European Commission | MSCA Long Term Fellowshipt ('Memento') | Marco Paoli |
| European Commission | ERC Starting Grant ('Emerg-Ant' 759817) | Antoine Wystrach |

| Funder | Grant reference number | Author |
| --- | --- | --- |
| European Commission | ERC Advanced Grant ('Cognibrains') | Martin Giurfa |

The funders had no role in study design, data collection and interpretation, or the decision to submit the work for publication.

### Author contributions

Marco Paoli, Conceptualization, Data curation, Software, Formal analysis, Validation, Investigation, Visualization, Methodology, Writing - original draft, Writing - review and editing; Antoine Wystrach, Conceptualization, Software, Methodology, Writing - review and editing; Brice Ronsin, Supervision, Methodology, Writing - review and editing; Martin Giurfa, Conceptualization, Supervision, Funding acquisition, Methodology, Project administration, Writing - review and editing

### Author ORCIDs

Marco Paoli ⓘ https://orcid.org/0000-0001-5672-1403
Martin Giurfa ⓘ https://orcid.org/0000-0001-7173-769X

Reviewer #1 (Public Review): https://doi.org/10.7554/eLife.93789.3.sa1
Author response https://doi.org/10.7554/eLife.93789.3.sa2

## Additional files

### Supplementary files

• MDAR checklist

### Data availability

An exemplary data subset and the main MatLab scripts used to model KC activity and MBON learning have been uploaded to GitHub (copy archived at *Paoli, 2024*). Calcium imaging dataset is available at Dryad.

The following dataset was generated:

| Author(s) | Year | Dataset title | Dataset URL | Database and Identifier |
| --- | --- | --- | --- | --- |
| Paoli M, Wystrach A, Ronsin B, Giurfa M | 2024 | Data from: Analysis of fast calcium dynamics of honey bee olfactory coding | http://dx.doi.org/10.5061/dryad.qbzkh18sc | Dryad Digital Repository, 10.5061/dryad.qbzkh18sc |

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
