## [Editor Report · eLife assessment]

How neural circuits represent sensory signals during and after stimulus presentation is a central question in neuroscience. Here, a model of the insect mushroom body, constructed from simple, known synaptic connectivity rules, is shown to **convincingly** explain stimulus discrimination and associative memory, even in the presence of variability in the input signals as experimentally measured from the antennal lobe of the honeybee. This **important** study makes testable predictions for the role of specific neurons in a neural circuit for associative memory, of relevance to any study of neural network design and operation.

---

## [Referee Report · Reviewer #1 (Public Review)]

This study by Paoli et al. used a resonant scanning multiphoton microscope to examine olfactory representation in the projection neurons (PNs) of the honeybee with improved temporal resolution. PNs were classified into 9 groups based on their response patterns. Authors found that excitatory repose in the PNs precedes the inhibitory responses for ~40ms, and ~50% of PN responses contain inhibitory components. They built the neural circuit model of the mushroom body (MB) with evolutionally conserved features such as sparse representation, global inhibition, and a plasticity rule. This MB model fed with the experimental data could reproduce a number of phenomena observed in experiments using bees and other insects, including dynamical representations of odor onset and offset by different populations of Kenyon cells, prolonged representations of after-smell, different levels of odor-specificity for early/delay conditioning, and shift of behavioral timing in delay conditioning. The trace conditioning was also tested experimentally, although bees did not shift the timing of PER response to the post-odor period as the model predicted. The experimental data and the model provide a solid basis for future studies.

---

## [Author Response]

The following is the authors’ response to the original reviews.

**Public Reviews:**

**Reviewer #1 (Public Review):**
This study by Paoli et al. used a resonant scanning multiphoton microscope to examine olfactory representation in the projection neurons (PNs) of the honeybee with improved temporal resolution. PNs were classified into 9 groups based on their response patterns. Authors found that excitatory repose in the PNs precedes the inhibitory responses for ~40ms, and ~50% of PN responses contain inhibitory components. They built the neural circuit model of the mushroom body (MB) with evolutionally conserved features such as sparse representation, global inhibition, and a plasticity rule. This MB model fed with the experimental data could reproduce a number of phenomena observed in experiments using bees and other insects, including dynamical representations of odor onset and offset by different populations of Kenyon cells, prolonged representations of after-smell, different levels of odorspecificity for early/delay conditioning, and shift of behavioral timing in delay conditioning. The trace conditioning was not modeled and tested experimentally. Also, the experimental result itself is largely confirmatory to preceding studies using other organisms. Nonetheless, the experimental data and the model provide a solid basis for future studies.

We thank the reviewer for summarizing the value of our study and recognizing its generality and significance. As suggested, in a revised version of the manuscript, we will discuss the implication of our approach for the context of trace conditioning. The model we presented hinges on the learning-induced plasticity of KC-to-MBON synapses recruited during the learning window (i.e., the simulated US arrival). In the case of trace conditioning, the model predicts that the time of the behavioral response time should match the expected US arrival. Contrary to this prediction, preliminary analyses on empirical measurements of PER latency upon trace conditioning indicate this is not the case. In a revised version of the manuscript, we will discuss the differences between the predictions of the model and the experimental observations in a trace conditioning paradigm.

**Reviewer #2 (Public Review):**
The study presented by Paoli et al. explores temporal aspects of neuronal encoding of odors and their perception, using bees as a general model for insects. The neuronal encoding of the presence of an odor is not a static representation; rather, its neuronal representation is partly encoded by the temporal order in which parallel olfactory pathways participate and are combined. This aspect is not novel, and its relevance in odor encoding and recognition has been discussed for more than the past 20 years.The temporal richness of the olfactory code and its significance have traditionally been driven by results obtained based on electrophysiological methods with temporal resolution, allowing the identification and timing of the action potentials in the different populations of neurons whose combination encodes the identity of an odor. On the other hand, optophysiological methods that enable spatial resolution and cell identification in odor coding lack the temporal resolution to appreciate the intricacies of olfactory code dynamics.(1) In this context, the main merit of Paoli et al.'s work is achieving an optical recording that allows for spatial registration of olfactory codes with greater temporal detail than the classical method and, at the same time, with greater sensitivity to measure inhibitions as part of the olfactory code.The work clearly demonstrates how the onset and offset of odor stimulation triggers a dynamic code at the level of the first interneurons of the olfactory system that changes at every moment as a natural consequence of the local inhibitory interactions within the first olfactory neuropil, the antennal lobe. This gives rise to the interesting theory that each combination of activated neurons along this temporal sequence corresponds to the perception of a different odor. The extent to which the corresponding postsynaptic layers integrate this temporal information to drive the perception of an odor, or whether this sequence is, in a sense, a journey through different perceptions, is challenging to address experimentally.In their work, the authors propose a computational approach and olfactory learning experiments in bees to address these questions and evaluate whether the sequence of combinations drives a sequence of different perceptions. In my view, it is a highly inspiring piece of work that still leaves several questions unanswered.

We thank the reviewer for considering that our work has an inspiring nature. Below we have tried to answer the questions raised by the following comments, and we will include part of these answers in the revised version of our manuscript.

(2) In my opinion, the detailed temporal profile of the response of projection neurons and their respective probabilities of occurrence provide valuable information for understanding odor coding at the level of neurons transferring information from the antennal lobes to the mushroom bodies. An analysis of these probabilities in each animal, rather than in the population of animals that were measured, would aid in better comprehending the encoding function of such temporal profiles. Being able to identify the involved glomeruli and understanding the extent to which the sequence of patterns and inhibitions is conserved for each odor across different animals, as it is well known for the initial excitatory burst of activity observed in previous studies without the fine temporal detail, would also be highly significant.

We thank the reviewer for recognizing the relevance of the findings in understanding the logic of olfactory coding. We agree about the importance of establishing if the different glomerular response profiles are evenly distributed across individuals or have individual biases. In the revised version of the manuscript, we will provide data on the distribution of response profiles for each animal and for different olfactory stimuli. Also, we fully agree on the importance of assessing to what extent such response profiles - largely determined by the local network of AL interneurons - are glomerulus-specific and conserved across individuals.

In my view, the computational approach serves as a useful tool to inspire future experiments; however, it appears somewhat simplistic in tackling the complexity of the subject. One question that I believe the researchers do not address is to what extent the inhibitions recorded in the projection neurons are integrated by the Kenyon cells and are functional for generating odor-specific patterns at that level.

The model we proposed represents, indeed, a simplification of olfactory signal processing throughout the honey bee olfactory circuit. Still, it shows that simple but realistic rules can be sufficient to grasp some fundamental aspects of olfactory coding. However, we agree with the reviewer and believe that such a minimalistic model can provide a basis for designing future experiments in which complexity can be increased by adding relevant features, such as the learning-induced plasticity of PN-to-KC synapses or the divergence of multiple PNs from the same glomerulus to different KCs.

Concerning the reviewer's question on the involvement of inhibitory inputs in generating odor-specific patterns at the level of the KCs, the short answer is yes, they contribute to the summed input of a target KC, thus to the odor representation. In designing the model, we considered that a given glomerulus provides maximal input at maximal excitation and minimal input (=0 input) at maximal inhibition. For this reason, an inhibited glomerulus contributes less (to KC action potential probability) than a glomerulus showing baseline activity. This, in turn, contributes less than an excited glomerulus. From the modeling point of view, normalizing the signal between 0 and 1 (i.e., setting minimal inhibition to 0 and maximal excitation to 1) would yield a similar result as with the current approach, where values range from -25% to +30% ΔF/F. We implement the model's description to clarify this point.

Lastly, the behavioral result indicating a difference in conditioned response latency after early or delayed learning protocol is interesting. However, it does not align with the expected time for the neuronal representation that was theoretically rewarded in the delayed protocol. This final result does not support the authors' interpretation regarding the existence of a smell and an after-smell as separate percepts that can serve as conditioned stimuli.

Considering that our odor stimulus lasted 5 seconds, glomerular activity is highly variable at odor onset (i.e., within the first 1s) because of short excitatory response profiles and the delayed and slower onset of inhibitory responses. After the initial phase, the neural representation of the stimulus becomes more stable. Consequently, a neural signature learned in the case of delay conditioning, i.e., with the US appearing towards the end of the olfactory stimulation (t = 4 - 5s), may present itself much earlier (t = 1.5s), triggering a behavioral response that largely anticipates the expected US arrival time.

In the model, we observe an early decrease in action potential probability even in the case of delay conditioning. This occurs because the synapses recruited during the last second of olfactory stimulation (within the learning window during which CS and US overlap) become inactive. Because odorant-induced activity recruits highly overlapping synaptic populations between 1.5 and 5 s from the onset, a learning-induced inactivation of part of these synapses will result in a reduced action-potential probability in the modeled MBON. Importantly, this event will not be governed by time but by the appearance of the learned synaptic configuration.

We will add a new section to the revised version of the manuscript to clarify this concept and perform further analyses to characterize the contribution of different response types to the modeled response latency.